# BAYESIAN RELATIONAL GENERATIVE MODEL FOR SCALABLE MULTI-MODAL LEARNING

## ABSTRACT

The study of complex systems requires the integration of multiple heterogeneous and high-dimensional data types (e.g. multi-omics). However, previous generative approaches for multi-modal inputs suffer from two shortcomings. First, they are not stochastic processes, leading to poor uncertainty estimations over their predictions. This is mostly due to the computationally intensive nature of traditional stochastic processes, such as Gaussian Processes (GPs), that makes their applicability limited in multi-modal learning frameworks. Second, they are not able to effectively approximate the joint posterior distribution of multi-modal data types with various missing patterns. More precisely, their model assumptions result in miscalibrated precisions and/or computational cost of sub-sampling procedure. In this paper, we propose a class of stochastic processes that learns a graph of dependencies between samples across multi-modal data types through adopting priors over the relational structure of the given data modalities. The dependency graph in our method, multi-modal Relational Neural Process (mRNP), not only posits distributions over the functions and naturally enables rapid adaptation to new observations by its predictive distribution, but also makes mRNP scalable to large datasets through mini-batch optimization. We also introduce mixture-of-graphs (MoG) in our model construction and show that it can address the aforementioned limitations in joint posterior approximation. Experiments on both toy regression and classification tasks using real-world datasets demonstrate the potential of mRNP for offering higher prediction accuracies as well as more robust uncertainty estimates compared to existing baselines and state-of-the-art methods.

## 1 INTRODUCTION

Many prominent methods for multi-modal learning (Argelaguet et al., 2018; Andrew et al., 2013; Klami et al., 2013; Zhao et al., 2016) are based on the canonical correlation analysis (CCA), which extracts shared components across multiple views. The main idea is that, given two vectors of random variables, the method finds the linear projections in a shared latent space, in which the projected vectors are maximally correlated (Thompson, 1984). This can help to understand the overall dependency structure between these two vectors. However, the classical CCA cannot handle non-linearity (Andrew et al., 2013) and suffers from a lack of probabilistic interpretation when applied to high dimensional data (Klami et al., 2013). To address these issues, probabilistic CCA (PCCA) has been proposed and extended to non-linear settings using kernel methods and neural networks (Bach & Jordan, 2005).

More recently, Wu & Goodman (2018) introduced the multimodal variational autoencoder (mVAE) that models the joint posterior as a product-of-experts (PoE) over the marginal posteriors, enabling cross-modal generation at test time without requiring additional inference networks and multi-stage training regimes. However, training a PoE is difficult and the following techniques are needed in order to ensure that the individual views are learnt faithfully; i) using artificial sub-sampling of the observed views (Wu & Goodman, 2018), ii): applying variants of contrastive divergence (Hinton, 2002), and iii): utilizing information bottleneck on the marginal representations of each view (Lee & Schaar, 2021). Furthermore, each expert holds the power of veto and low density of only one marginal posteriors among a given set of observations leads to the low density of joint distribution (Shi et al., 2019). In the case of Gaussian experts, different levels of complexity of modalities or

sub-optimal initialisation might result in the miscalibrated precisions that potentially lead to a biased overall mean prediction (Shi et al., 2019).

Despite the success of the existing multi-modal generative methods, they suffer from inducing stochasticity in their inferred functions. This leads to; i) poor adaptation to new observations[1], ii) inefficient generalizability in few-shot settings and, iii) inaccurate uncertainty estimation, all of which are important in many safety critical applications such as healthcare. On the other hand, stochastic processes, such as Gaussian processes (GPs), are exchangeable models that posit distributions over possible functions, and are updated in light of data through the probabilistic inference. Despite these advantages, GPs are computationally expensive and their underlying model is not flexible for high-dimensional inputs, making them infeasible for multi-modal learning settings with multiple heterogeneous and high-dimensional data types.

Apart from the aforementioned issues, existing generative models for multi-modal learning focus on latent representation, but do not fully incorporate the label information. In order to facilitate predictions on the target labels, they need to apply two-step procedures, in which the shared latent spaces are used for downstream classification. While information relevant to the reconstruction of views will be well-captured in the shared representations, discriminative information relevant to the target task may be discarded, leading to poor prediction performance of such models (Lee & Schaar, 2021). Furthermore, the shared representations of these methods in high-dimensional space are often difficult to interpret and need an extra downstream analyses such as PCA or t-SNE (Van Der Maaten, 2014) that may cause biased results (Lötsch & Ultsch, 2020).

In this paper, instead of following the existing multi-modal generative methods that rely on deterministic predictive functions, we propose the first multi-modal stochastic processes family that learns distributions over functions for inputs with any missing patterns. Like GPs, it naturally provides robust uncertainty estimates and can encode inductive biases. Unlike GPs, though, mRNP is computationally efficient during training and evaluation since it can take advantage of mini-batch optimization, and learns to adapt their priors to data. The inferred dependency graph in our model construction can be used to visualise the high dimensional latent representation, obviating the need for *ad-hoc* post-processing steps as required in most of the multi-modal learning methods.

The presented work makes four major contributions: 1) We develop a novel multi-modal Relational Neural Process, mRNP, that defines a distribution over functions of multiple data types by employing local latent variables, and learns a dependency structure among the samples of the given modalities. 2) We theoretically prove exchangeability and consistency of mRNP, two necessary conditions that have to be satisfied during the construction of such a model, showing that mRNP is a valid stochastic process. 3) We further show that the local latent variable structure in mRNP is able to encode inductive biases and demonstrate this by designing an mRNP model that behaves similarly to a GP with an RBF kernel (an ablation study). 4) We introduce mixture-of-graphs (MoG) in our model construction that can address the issues such as computational complexity and miscalibrated precisions observed in the previous approaches in multi-modal learning.

## 2 METHOD

We propose a new graph-structured multi-modal learning method, referred as multi-modal Relational Neural Process (mRNP), that combines the benefits of neural networks with that of stochastic processes. In the supervised multi-modal setting, our dataset contains sets of feature-label pairs, $\{\bar{\mathbf{x}}, y\}$ from $|\mathcal{V}|$ different modalities with various missing patterns, where $|\mathcal{V}|$ is number of views, $\bar{\mathbf{x}} = \{\mathbf{x}_v | v^{\text{th}} \text{ modality present}\}_{v \in \mathcal{V}} \in \bar{\mathcal{X}}$ is the input modalities and $y \in \mathcal{Y}$ is the given label. One of the key motivation of mRNP is that it gives us the ability to define a distribution over functions rather than learning a single static function when fitting the data. In this framework, inspired by the idea of inducing points in sparse Gaussian processes (Titsias, 2009; Damianou & Lawrence, 2013), we first select a *reference* set of samples that consists of functions $f : \bar{\mathbf{x}} \to y$ that are sampled from some underlying distribution $\mathcal{H}$. We then establish a probability distribution over $f_h$ around those samples. We define the *reference set*, $\mathcal{X}_R = \{\mathbf{X}_{R,1}, \dots, \mathbf{X}_{R,V}\} \subset \bar{\mathcal{X}}$, where $\mathbf{X}_{R,v} \in \mathbb{R}^{N_v \times D_v}$, and

---

[1] *Rapid adaption to new observations* is what distinguishes a model that can learn a whole distribution over functions by seeing different functions in each batch from the one that learns a single underlying function by observing several batches during the training. At the test time, the former will narrow down what the current function is by seeing a few context observations.

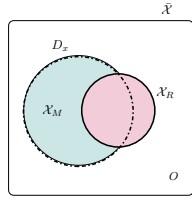
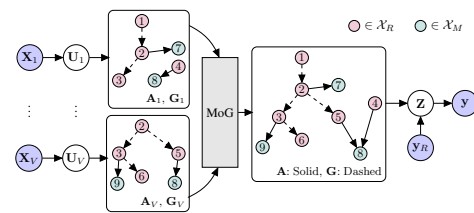

Figure 1: **Left:** Venn diagram of the sets where $\mathcal{X}_R$ and $\mathcal{X}_M$ are colored as pink and teal, respectively; and the samples enclosed in the dashed circle are $D_x$. $O$ is the complement of $\mathcal{X}_R$ and $\bar{\mathcal{X}} = O \cup \mathcal{X}_R$. **Right:** Graphical model for the proposed mRNP.

$N_v$ and $D_v$ are the number of samples and dimension of features in the domain $v$, respectively. We also define $O = \bar{\mathcal{X}} \setminus \mathcal{X}_R$, which is the set of all possible samples that are not in $\mathcal{X}_R$. Please note that two sets $\mathcal{X}_R$ and $O$ includes both labeled and unlabeled samples. We also denote our labeled samples as $D_x = \{\bar{\mathbf{x}}_1, \ldots, \bar{\mathbf{x}}_N\}$, a set of any finite random set from $\bar{\mathcal{X}}$, as well as defining sets $\mathcal{X}_M = D_x \setminus \mathcal{X}_R$ and $\mathcal{X}_B = \mathcal{X}_R \cup \mathcal{X}_M$. The Venn diagram of the sets are provided in Figure 1. We infer the parameters of our probabilistic model using variational inference. In the following, we first introduce each of the latent variables in our model as well as their corresponding prior and posterior distributions, and then prove that it corresponds to an infinitely exchangeable stochastic process. The graphical model of mRNP is illustrated in Figure 1.

**Modality-specific latent spaces.** The first step is to construct modality-specific latent representations, denoted by $\mathcal{U} = \{\mathbf{U}_v\}_{v \in \mathcal{V}}$, a set of $N_v \times D_u$ matrices, which independently embed samples of each modality (i.e. $N_v$) into a $D_u$ dimensional latent space as follows:

$$\int p_\theta(\mathcal{U}_B, \mathcal{X}_B) \, \mathrm{d}\mathcal{X} = \prod_{v \in \mathcal{V}} \int p_\theta(\mathbf{U}_{B,v}, \mathbf{X}_{B,v}) \, \mathrm{d}\mathbf{X}_{B,v} = \prod_{v \in \mathcal{V}} \int p_\theta(\mathbf{U}_{B,v} \,|\, \mathbf{X}_{B,v}) \, p(\mathbf{X}_{B,v}) \, \mathrm{d}\mathbf{X}_{B,v}.$$

We further parameterize the distribution over the samples $\mathbf{u}_{i,v}$ independently:

$$p_\theta(\mathbf{U}_{B,v} \,|\, \mathbf{X}_{B,v}) = \prod_{i \in B_v} p_\theta(\mathbf{u}_{i,v} \,|\, \mathbf{x}_{i,v}), \tag{1}$$

where $p_\theta(\mathbf{u}_{i,v} \,|\, \mathbf{x}_{i,v})$ can be any distribution that is derived from the observed input data. For simplicity, we use a diagonal Gaussian, the parameters of which are a function of the observed input. More specifically, we use two functions denoted by $\varphi_v^{\mathrm{emb},\mu}(\mathbf{X}_{B,v})$ and $\varphi_v^{\mathrm{emb},\sigma}(\mathbf{X}_{B,v})$ to infer the mean and variance of the distribution for each modality. Depending on the nature of the modality, these functions can be implemented using any highly expressive functions such as many variants of neural networks (NNs) and graph NNs.

**A directed graph among the reference samples.** The next step is to construct a directed graph across the samples in $\mathcal{X}_R$ using different modalities. Given the latent embedding $\mathcal{U}_R$, we first construct a set of directed acyclic graphs (DAGs), $\mathcal{G} = \{\mathbf{G}_v\}_{v \in \mathcal{V}}$, where $\mathbf{G}_v$ is a random binary adjacency matrix between reference samples present in the $v^{\mathrm{th}}$ modality. Then we combine the graphs $\mathbf{G}_v$ from multiple views to construct a common graph $\mathbf{G}$. Inspired by the concept of stochastic orderings (Shaked & Shanthikumar, 2007), we impose a topological ordering over the vectors in $\mathcal{U}_R$ to avoid cycles in each individual $\mathbf{G}_v$. The distribution of the adjacency matrices are defined as follows:

$$p(\mathbf{G}_v \,|\, \mathbf{U}_{R,v}) = \prod_{i \in R_v} \prod_{j \in R_v, j \neq i} \mathrm{Bernoulli}\left(\mathbf{G}_{ij,v} \,|\, \mathbb{I}[t(\mathbf{u}_{i,v}) > t(\mathbf{u}_{j,v})] \, \varphi^{\mathrm{sim}}(\mathbf{u}_{i,v}, \mathbf{u}_{j,v})\right). \tag{2}$$

Similar to Louizos et al. (2019), we employ a parameter free scalar projection $t(\mathbf{u}_{i,v}) = \sum_k t_k(\mathbf{u}_{ik,v})$, where $t_k(\cdot)$ is a monotonic function. In practice, we use the log cumulative distribution function of a standard normal distribution. Given the graphs $\mathbf{G}_v$, we construct a directed graph $\mathbf{G}$, in which the weight of each edge is average of its weights in different modalities, i.e. $\mathbf{G} = \sum_{v \in \mathcal{V}} \beta_v \mathbf{G}_v / \sum_{v \in \mathcal{V}} \mathbf{m}_v \mathbf{m}_v^T$, where $\mathbf{m}_v$ is a binary vector and $\mathbf{m}_{i,v} = 1$ when sample $i$ is observed in the modality $v$, and $\beta_v = 1/|V|$, assuming that the all modalities are equally important. The proposed formulation, referred as **mixture-of-graphs (MoG)**, is used to learn the shared structured representation of multi-modal inputs by using the graphs of individual modalities.

**A relational graph of dependencies from reference samples to $\mathcal{X}_M$.** In addition to the directed graph $\mathbf{G}$ used for the reference samples, we apply MoG to construct a bipartite graph

$\mathbf{A} = \sum_{v \in \mathcal{V}} \alpha_v \mathbf{A}_v$ that is a bi-adjacency matrix from $\mathcal{X}_R$ to $\mathcal{X}_M$. Given the latent embedding $\mathcal{U}_B$ as described previously, we model the elements of each individual bi-adjacency matrix by Bernoulli distribution as follows:

$$p(\mathbf{A}_v \,|\, U_{M,v}\,,\, U_{R,v}) = \prod_{i \in M_v} \prod_{j \in R_v} \text{Bernoulli}\left(\mathbf{A}_{ij} \,|\, \varphi^{\text{sim}}(\mathbf{u}_{i,v}\,,\, \mathbf{u}_{j,v})\right), \qquad (3)$$

where $\varphi^{\text{sim}}(\cdot, \cdot)$ is a score function measuring the similarity between the latent representations of the input samples in each modality $v$, and $\alpha_v = 1/|V|$, assuming the all modalities equally important. Depending on the desired relational inductive biases, we can appropriately define the function $\varphi^{\text{sim}}(\cdot, \cdot)$. The Bernoulli-Poisson link $\varphi^{\text{sim}}(\mathbf{u}_{i,v}, \mathbf{u}_{j,v}) = 1 - \exp(-\sum_{k=1}^{D_u} \tau_k u_{ik,v} u_{jk,v})$ and $\varphi^{\text{sim}}(\mathbf{u}_{i,v}, \mathbf{u}_{j,v}) = \exp(-\frac{\tau}{2} ||\mathbf{u}_{i,v} - \mathbf{u}_{j,v}||^2)$ are two examples of such potential score functions. Please note that we use the latter one throughout this paper.

For both $\mathcal{A} = \{\mathbf{A}_v\}_{v \in \mathcal{V}}$ and $\mathcal{G}$, we use the concrete relaxation (Maddison et al., 2016; Gal et al., 2017) during training while we sample from Bernoulli distributions in the testing phase.

**A shared latent variable.** Having obtained the sample representations $\mathcal{U}$ and the dependency graphs $\mathbf{A}$ and $\mathbf{G}$, we construct a shared latent variable, denoted by $N \times D_z$ matrix $\mathbf{Z}$, which can be used to predict target variable distributions $y_i$. We parameterize the distributions of the predictive targets as follows:

$$\int p_\theta(\mathbf{y}_B, \mathbf{Z}_B \,|\, \mathbf{A}, \mathbf{G}, \mathcal{X}_R) \mathrm{d}\mathbf{Z}_B = \int \int p_\theta(\mathbf{y}_R, \mathbf{Z}_R \,|\, \mathbf{G}, \mathcal{X}_R)\, p_\theta(\mathbf{y}_M, \mathbf{Z}_M \,|\, \mathbf{A}, \mathcal{X}_R, \mathbf{y}_R)\, \mathrm{d}\mathbf{Z}_R\, \mathrm{d}\mathbf{Z}_M$$

$$= \prod_{i \in R} \int p_\theta(\mathbf{z}_i \,|\, \text{par}_{\mathbf{G}_i}(\mathcal{X}_R, \mathbf{y}_R))\, p_\theta(y_i \,|\, \mathbf{z}_i)\, \mathrm{d}\mathbf{z}_i \prod_{j \in M} \int p_\theta(\mathbf{z}_j \,|\, \text{par}_{\mathbf{A}_j}(\mathcal{X}_R, \mathbf{y}_R))\, p_\theta(y_j \,|\, \mathbf{z}_j)\, \mathrm{d}\mathbf{z}_j,$$

where $\text{par}_{\mathbf{G}_i}(\cdot)$, $\text{par}_{\mathbf{A}_j}(\cdot)$ are functions that return the parents of the points $i$ and $j$ according to $\mathbf{G}$ and $\mathbf{A}$, respectively. We summarize the information from the parent samples and their targets in $\mathcal{X}_R$ through the local latent variable $\mathbf{z}_i$, allowing the distribution $\mathbf{y}_i$ to be explicitly dependent on the available graph dependency structures $\mathbf{A}$ and $\mathbf{G}$. We set the distribution over $\mathbf{z}_i$ as an independent Gaussian distribution whose parameters are a function of either $\mathbf{A}$ or $\mathbf{G}$ although any distribution with a permutation invariant probability density with respect to the parents can be used. More formally, the following distribution is adopted:

$$p_\theta(\mathbf{z}_i \,|\, \text{par}_{\mathbf{A}_i}(\mathcal{X}_R, \mathbf{y}_R)) = \prod_{k=1}^{D_z} p_\theta(z_{ik} \,|\, \text{par}_{\mathbf{A}_i}(\mathcal{X}_R, \mathbf{y}_R)) = \prod_{k=1}^{D_z} \mathcal{N}(\mu_{ik}^{\text{prior}}, \sigma_{ik}^{\text{prior}}), \qquad (4)$$

$$\mu_{ik}^{\text{prior}} = c_i \sum_{j \in \mathcal{X}_R} \mathbf{A}_{ij} \varphi_k^{\text{prior},\mu}(\overline{\mathbf{x}}_{j,R}\,,\, y_{j,R}), \quad \sigma_{ik}^{\text{prior}} = \exp(c_i \sum_{j \in \mathcal{X}_R} \mathbf{A}_{ij} \varphi_k^{\text{prior},\sigma}(\overline{\mathbf{x}}_{j,R}\,,\, y_{j,R})),$$

where $\overline{\mathbf{x}}_{j,R} = \{\mathbf{x}_{j,R,v}\}_{v \in \mathcal{V}}$, $\varphi^{\text{prior},\mu}$ and $\varphi^{\text{prior},\sigma}$ are transform functions with a co-domain $\mathbb{R}^{|z|}$, and $c_i = (\sum_j \mathbf{A}_{ij} + \epsilon)^{-1}$ is a normalization constant. Given input $\{\mathbf{X}_{B,v}\}_{v \in \mathcal{V}}$, we factorize the variational posterior distribution as

$$q_\phi(\mathbf{Z} \,|\, \mathcal{X}_B) = \prod_{i \in B} q_\phi(\mathbf{z}_i \,|\, \overline{\mathbf{x}}_{i,B}), \qquad (5)$$

In practice, we also parameterize the priors over the latent $\mathbf{z}_i$ in terms of the posterior distribution of the reference samples, i.e. $q_\phi(\mathbf{z}_i \,|\, \mathbf{x}_{i,R})$. More precisely, we define $\varphi^{\text{prior},\mu}$ and $\varphi^{\text{prior},\sigma}$ in equation (4) as $\varphi^{\text{post},\mu}(\mathbf{x}_{i,R}) + \varphi_{\text{label}}^{\text{emb},\mu}(y_{i,R})$ and $\varphi^{\text{post},\sigma}(\mathbf{x}_{i,R}) + \varphi_{\text{label}}^{\text{emb},\sigma}(y_{i,R})$, respectively; where $\varphi^{\text{post}}$ and $\varphi_{\text{label}}^{\text{emb}}$ provide the means and variances of $q_\phi(\mathbf{z}_i \,|\, \mathbf{x}_{i,R})$ and the linear embedding of the labels of reference samples, respectively.

We should point out that in this formulation, the prediction of $y_i$ indirectly depends on the input samples $\overline{\mathbf{x}}_i$ through the graphs $\mathbf{G}$ and $\mathbf{A}$, which are functions of $\overline{\mathbf{u}}_i$. This leads to an uninformative standard normal prior over $\mathbf{z}_i$ for the samples with very small probability of being connected to the reference set through $\mathbf{A}$, and hence the prediction of $y_i$ will be constant. This can be seen as encoding inductive bias similar to a GP with an RBF kernel.

**MoG vs PoE.** PoE is a prime choice to learn the variational joint posteriors; it can learn under any combination of missing modalities (Shi et al., 2019). Despite this advantage, it also has two main limitations: 1) the underlying model suffers from the overconfident experts since experts with

greater precision will have more influence over the combined prediction than experts with lower precision; 2) training and inference can be costly due to its artificial sub-sampling procedure. Given the aforementioned limitations of PoE, we propose to build a graph of dependencies among local latent variables as a mixture of modality-specific graphs, i.e. *mixture-of-graph* (MoG), and then parameterize the joint latent space $\mathbf{Z}$ conditioned on those graph. Unlike PoE, MoG effectively spreads its density over all the individual modalities by imposing a weight to each individual graphs. When prior knowledge is available, terms $\alpha_v$ and $\beta_v$ in the MoG construction can be used to encourage certain modalities, which is the case in multi-omics data integration where individual modalities often contain complementary information of target task. A more detailed discussion can be found in Section D of the supplement.

We also should point out that mRNP is not a graph learning model. Rather, we propose a novel multi-view NPs that is able to solve a pitfall in utilizing stochastic processes for multi-view setting by learning a graph of dependencies. We can also consider graph learning in mRNP as a kind of cross-attention in the form of a dependency graph among local latent variables. From this viewpoint, mRNP is learning attention mechanism to combine modalities. The current framework in which mRNP learns the individual graphs might not be computationally efficient, especially if training set is large. We may improve the scalability issue of graph construction part of our mRNP by e.g. considering hierarchical structure similar to hFGW (Xu et al., 2020). We leave this for a future study.

**Overall likelihood and learning.** Putting everything together, the marginal likelihood is

$$p_\theta(\mathbf{y}_B \,|\, \mathcal{X}_B) = \sum_{\mathbf{G},\mathbf{A}} \int \prod_{v \in \mathcal{V}} p_\theta(\mathbf{U}_{B,v} \,|\, \mathbf{X}_{B,v}) \, p(\mathbf{G}, \mathbf{A} \,|\, \mathcal{U}_B) \, p_\theta(\mathbf{y}_B, \mathbf{Z}_B \,|\, \mathbf{A}, \mathbf{G}, \mathcal{X}_R) \, \mathrm{d}\mathcal{U}_B \, \mathrm{d}\mathbf{Z}_B. \quad (6)$$

We should point out that, for the sake of simplicity, we choose the reference set to be part of the training set $\mathcal{D}_x$ throughout this paper. More specifically, we assume $\mathcal{X}_B = \mathcal{D}_x$. In case of existing reference samples that are not part of the training set, we need to marginalize out over them in equation (6) in addition to the marginalizations over the latent variables and structured graphs. We can also use the marginalization technique to incorporate unlabelled data in order to learn a better representation and/or to impute the missing labels. We leave this for a future study.

We deploy variational inference to optimize the model parameters $\theta$ and variational parameters $\phi$ by minimizing the following derived Evidence Lower Bound (ELBO):

$$\mathcal{L} = \mathbb{E}_{q_\phi(\mathbf{Z}_R \,|\, \mathcal{X}_R) \, p(\mathbf{G} \,|\, \mathcal{U}_R) \, \prod_{v \in \mathcal{V}} p_\theta(\mathbf{U}_{R,v} \,|\, \mathbf{X}_{R,v})} \left[ \log p_\theta(\mathbf{y}_R, \mathbf{Z}_R \,|\, \mathbf{G}, \mathcal{X}_R) - \log q_\phi(\mathbf{Z}_R \,|\, \mathcal{X}_R) \right] +$$

$$\mathbb{E}_{q_\phi(\mathbf{Z}_M \,|\, \mathcal{X}_M) \, p(\mathbf{A} \,|\, \mathcal{U}_B) \, \prod_{v \in \mathcal{V}} p_\theta(\mathbf{U}_{B,v} \,|\, \mathbf{X}_{B,v})} [\log p_\theta(\mathbf{y}_M \,|\, \mathbf{Z}_M) +$$

$$\log p_\theta(\mathbf{Z}_M \,|\, \mathrm{par}_\mathbf{A}(\mathcal{X}_R, \mathbf{y}_R)) - \log q_\phi(\mathbf{Z}_M \,|\, \mathcal{X}_M)].$$

Please note that we can use mini-batches for the second expectation term where the size of the batches scale according to the size of the reference set $\mathcal{X}_R$. Due to the DAG structure in the first expectation term, we cannot decompose it to independent sums.

**Predictive distribution.** After optimizing the model parameters, we derive the predictive distribution for unseen samples $\{\mathbf{x}_v^*\}_{v \in \mathcal{V}}$ with missing modalities based on Bayes theorem as follows:

$$p_\theta(y^* \,|\, \{\mathbf{x}_v^*\}_{v \in \mathcal{V}}, \mathcal{X}_B, \mathbf{y}_B) = \sum_{\mathbf{a}^*} \int \prod_{v \in \mathcal{V}} p_\theta(\mathbf{u}_v^*, \mathbf{U}_{R,v} \,|\, \mathbf{x}_v^*, \mathbf{X}_{R,v})$$

$$p(\mathbf{a}^* \,|\, \{\mathbf{u}_v^*\}_{v \in \mathcal{V}}, \mathcal{U}_R) \, p_\theta(\mathbf{z}^* \,|\, \mathrm{par}_{\mathbf{a}^*}(\mathcal{X}_R, \mathbf{y}_R)) \, p_\theta(y^* \,|\, \mathbf{z}^*) \, \mathrm{d}\mathcal{U}_R \, \mathrm{d}\mathbf{u}_{v \in \mathcal{V}}^* \, \mathrm{d}\mathbf{z}^*,$$

where $\mathbf{u}_v^*$ is the embedding representation of the observed modality $\mathbf{x}_v^*$ through the neural network $\varphi_v^{\mathrm{emb}}$ and $\mathbf{a}^*$ is a binary vector similar to a row of $\mathbf{A}$ that denotes reference parents of the new samples. This is similar to the predictive distribution in few-shot learning (Sung et al., 2018).

**Proposition I.** mRNP corresponds to Bayesian models as the distributions defined in Equation (6) are valid, permutation invariant stochastic processes.

*proof.* The proof is deferred to the supplement.

## 3 RELATED WORKS

**Multi-modal VAE.** There has been a long line of research in using variants of VAEs to explicitly model the joint distribution over latents and data (Suzuki et al., 2016; Vedantam et al., 2017; Wu

& Goodman, 2018; Shi et al., 2019). Most of these works are not able to efficiently explore more than two modalities and/or suffer from multi-stage training. Using PoE, mVAE (Wu & Goodman, 2018) addresses the aforementioned problems and learns a joint embedding under any combination of missing modalities. However, 1) it suffers from the miscalibration of the precision of the experts due to differences in complexity of input modalities, and 2) given a complete dataset with no missing modalities, it requires artificial sub-sampling in order to faithfully learn the individual modalities.

Closely related to the mVAE, DeepIMV (Lee & Schaar, 2021) applies PoE in a supervised framework for multi-omics data integration with missing views. Our mRNP is similar to this method in terms of using supervised framework. But, unlike mRNP, DeepIMV predictive function is deterministic and therefore is not suitable for out-of-distribution prediction and/or few-shot settings. It also includes a set of $|\mathcal{V}|$ view-specific predictors in its construction and apply information bottleneck (IB) principle on the marginal representations of individual views in order to allow each encoder to build view-specific expertise for predicting the target. Depending on the number of input modalities, computational complexity might be an issue.

**Neural process (NP).** NPs (Garnelo et al., 2018b;a) have been recently proposed to learn an approximation of a stochastic process by using a neural network-based formulation. Traditional stochastic processes such as GPs are usually computationally expensive and the available kernels are usually restricted in their functional form and are not very flexible for high dimensional problems (Garnelo et al., 2018b). On the other hand, NPs are data-driven models that impose stochasticity in the function realizations and are able to define a flexible class of stochastic processes well suited for highly non-trivial functions(Lee et al., 2020). The main idea behind NPs is to define explicit global latent variables in their construction which can capture functional uncertainty(Garnelo et al., 2018b). Closely related to our mRNP, Louizos et al. (2019) discards the idea of the global latent variables and instead builds a graph of dependencies among local latent variables, making it more suitable for modeling high-dimensional data. There are other NP variants (Kim et al., 2019; Lee et al., 2020) to improve the performance of NPs. However, none of these works addresses multi-modal learning problem. To the best of our knowledge, mRNP is the first NP-based model to explore multi-modal in a computationally efficient way.

## 4 EXPERIMENTS

We evaluate the effectiveness of mRNP based on two different multi-modal experiments: 1) On two real-world datasets, we measure the performance of mRNP by using its prediction scores and the quality of uncertainty quantification in multi-modal classification tasks under any combination of missing modalities; 2) We investigate the inductive biases that we can encode in mRNP in an ablation study by visualizing the predictive distributions in two toy examples of 1-dimensional multi-modal regression tasks. We compare its performance with those of three state-of-the-art methods; MOFA (Argelaguet et al., 2018), mVAE (Wu & Goodman, 2018), and DeepIMV (Lee & Schaar, 2021). We should emphasize that MOFA and mVAE models are unsupervised embedding models and suffer from task specific training. Specifically, these models derive low-dimensional (non-)linear embedding of the input samples, and hence a classification model has to be trained for a downstream analysis task. We should also point out that only autoencoder-based methods are able to be trained using the observed samples, for which the optimized parameters can be used for unseen samples. So, for MOFA, we need to combine training and test data in order to learn the low-dimensional embedding.

We implemented our model in PyTorch (Paszke et al., 2019). Throughout the classification experiments, we use two fully connected layers with ReLu as the activation function at each hidden layer to obtain an intermediate hidden representation $\mathbf{h}_v$ of the inputs $\mathbf{x}_v$, and then parameterize a linear output layer to learn the parameters of $p_\theta(\mathbf{u}_v \mid \mathbf{x}_v)$, which we consider as Gaussian. We also factorize the posterior $q_\phi(\mathbf{Z} \mid \mathcal{X}_B) = \prod_{i \in B} \left( p(\mathbf{z}_i) \prod_{v \in \mathcal{V}} q_\phi^{(v)}(\mathbf{z}_i \mid \mathbf{x}_{i,B,v}) \right)$, where $p(\mathbf{z}_i) = \mathcal{N}(\boldsymbol{\mu}_0, \boldsymbol{\Sigma}_0)$, and $q_\phi^{(v)}(\mathbf{z}_i \mid \mathbf{x}_{i,B,v}) = \mathcal{N}(\boldsymbol{\mu}_v, \boldsymbol{\Sigma}_v)$ is the underlying inference network for the modality $v$. We use a linear layer as an encoder for each modality. For fair comparison, the results of the competing methods are obtained based on their original implementations with the same architectures as mRNP. For the regression tasks, we used one fully connected layer with ReLu as the activation function to transfer $\mathbf{x}_v$ to $\mathbf{h}_v$, followed by two separate linear layers to learn the parameters of $p_\theta(\mathbf{u}_v \mid \mathbf{x}_v)$, and $q_\phi(\mathbf{z}_v \mid \mathcal{X})$, respectively. We train mRNP for a maximum of 50 (5000) epochs using Adam (Kingma

Table 1: Accuracy and uncertainty on HW from 100 posterior predictive samples. The first column is the average predictive entropy whereas for the o.o.d. datasets the second is the AUC/AP and for the in-distribution it is the test accuracy in %.

|  | MOFA | mVAE | DeepIMV | **mRNP** |
|---|---|---|---|---|
| HW | NA / 78.72±0.0 | 4.07 / 66.98±3.17 | 2.28 / 72.07±2.36 | **0.40 / 82.50±0.8** |
| Gaussian | NA | 4.04 / 50.19/58.50 | 2.30 / 88.83 / 87.77 | **1.40 / 92.27 / 89.77** |
| Uniform | NA | 4.00 / 51.54/61.78 | 2.30 / 94.81 / 93.22 | **1.63 / 97.96 / 94.15** |
| Poisson | NA | 3.48 / 43.39/47.52 | 2.26 / 27.59 / 36.54 | **1.86 / 95.58 / 96.16** |
| Average | NA | 3.84±0.3 / 48.37±4.5 / 61.25±12.6 | 2.29±0.0 / 70.4±37.2/ 72.5±31.3 | **1.6±0.2 / 95.3±2.8 / 93.4±3.2** |

& Ba, 2014) with a learning rate of 0.001 for classification (regression). More implementation details are included in the supplement.

## 4.1 PREDICTION PERFORMANCE AND UNCERTAINTY QUALITY

For the classification task, we consider two publicly available real-world, multi-modal datasets; the HandWriting (HW) (Dua & Graff, 2017) and the Cancer Cell Line Encyclopedia (CCLE) (Barretina et al., 2012). Throughout the experiments, the dimensions $\{D_u, D_z, D_h\}$ are $\{32, 64, 200\}$ and $\{32, 32, 32\}$ for HW and CCLE, respectively. We use 300 (150) random samples from the training set as $\mathcal{X}_R$ for HW (CCLE). We train mRNP for 50 (10) epochs for HW (CCLE) and use the validation set for the early stopping. All of our results are averaged over 100 runs, ten randomly drawn dataset splits and ten runs with different random seeds on each split.

In order to capture uncertainty, we choose the predictive entropy that combines both epistemic and aleatoric uncertainties (Mukhoti & Gal, 2018). In particular, we assume that epistemic uncertainty of our Bayesian model will increase in areas where we have no data, i.e. out of distribution (o.o.d.) datasets, while the predictive entropy of in-distribution can be used to compare different models in terms of combination of both uncertainties. In order to determine whether a sample is in or out of distribution based on its predictive entropy, we also report the area under both the receiver operating characteristic (ROC) and precision-recall (PR) curves. We should point out that, by using ROC and PR, we can make sure that the improvement in predictive entropy is not due to a trivial model/learning since the model must have low entropy on the in-distribution test set but high entropy on the o.o.d. datasets in order to perform well in both ROC and PR. For the HW dataset, we consider Gaussian $\mathcal{N}(0, 1)$, uniform $U[0, 1]$, and $\mathrm{Poisson}(\lambda = \max(\mathbf{x}))$ noises as o.o.d. datasets. For CCLE, we use three real-word cancer datasets COAD, KIRC, and SKCM from The Cancer Genome Atlas (TCGA) (Tomczak et al., 2015) as well as Poisson noise as o.o.d datasets.

**Multi-view handwriting data with missing views.** First, we consider the classification of multi-view handwriting data. This dataset consists of features of handwritten numerals, i.e. $\{0, \ldots, 9\}$, extracted from a collection of Dutch utility maps (van Breukelen et al., 1998). The dataset includes 76 Fourier coefficients of the character shapes, 47 Zernike moments and 6 morphological features for a total of 2,000 samples (200 samples per class). We follow the same data pre-processing as in Lee & Schaar (2021), and choose 20% and 16% of samples as test and validation sets.

The summary of the results are reported in Table 1. The proposed method outperforms the state-of-the-art (SOTA) methods by a significant margin in terms of both classification and o.o.d prediction accuracies. Our Bayesian mRNP model performs significantly better than DeepIMV, which is the only supervised baseline, in terms of uncertainty quantification. This might be

Table 2: Comparison of the joint representation learning of HW data.

| Method | mVAE | DeepIMV | **mRNP** |
|---|---|---|---|
| Accuracy | 66.9 | 67.5 | **0.78** |
| NMI | 0.2 | 0.60 | 0.78 |

due to the fact that the softmax entropy in DeepIMV is only able to capture aleatoric uncertainty while features with high epistemic uncertainty are ignored, resulting in a relatively high number of inaccurate but certain samples. mRNP, on the other hand, captures both aleatoric and epistemic uncertainties.

We also compare the shared (structured) representations of different modalities from mRNP with the joint latent representations learned by DeepIMV and mVAE. Figure 2 shows an example of the graph $\mathbf{G}$ that mRNP learns, as well as PCA projections of latent representations from DeepIMV and mVAE. First, it shows that mRNP learns a meaningful $\mathbf{G}$ by connecting samples with same class to each other. Second, a few samples that are not in the same community as their labeled class, have

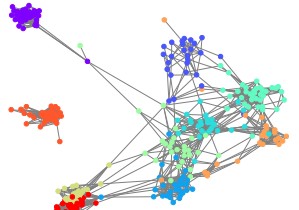 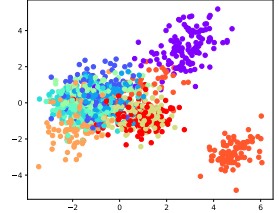 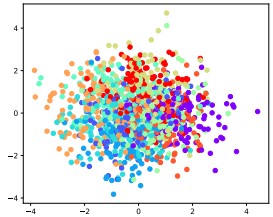

Figure 2: Comparison of the relational graph **G** on HW, inferred using mRNP (left) with PCA projections of latent representations from DeepIMV (middle) and mVAE (right). Nodes' colors refer to their class labels, i.e 1-10.

lower degrees compared to the other nodes in the same community. This indicates that the model is less confident about the class of such a sample. The proposed mRNP provides more interpretable shared representation than traditional non-structured representations while obviating the need for post-processing step to visualize a high-dimensional embedding space in a 2- or 3-dimensional one. Furthermore, we evaluate the joint representations of these models on the HW dataset through two downstream tasks: clustering and classification. For the first task, we apply K-means clustering and for the second one we train a linear SVM. The normalized mutual information (NMI) and classification accuracies are reported in Table 2.

We further perform an ablation study on HW dataset and provide some additional insights regarding the sensitivity of mRNP to the number of reference samples. Table 3 shows the test accuracy, entropy, o.o.d entropy and AUC for o.o.d prediction. Note that the mean and standard error across all of the o.o.d. datasets are reported for both o.o.d. entropy and AUC. We observe that the choice of samples as a reference set is important for mRNP

Table 3: Comparison of test accuracy and uncertainty quality in HW for different number of samples in $\mathcal{X}_R$.

| # of $R$ | Accuracy | entropy | o.o.d. entropy | AUC |
|---|---|---|---|---|
| 50 | $48.35 \pm 15.8$ | 1.17 | $1.8 \pm 0.26$ | $81.91 \pm 12.2$ |
| 100 | $80.75 \pm 1.7$ | 0.41 | $1.4 \pm 0.15$ | $94.04 \pm 2.0$ |
| 200 | $82.40 \pm 1.3$ | 0.37 | $1.5 \pm 0.26$ | $95.13 \pm 3.2$ |
| 300 | $\mathbf{82.52 \pm 1.1}$ | 0.40 | $1.6 \pm 0.23$ | $\mathbf{95.27 \pm 2.85}$ |
| 400 | $81.47 \pm 1.5$ | 0.39 | $1.5 \pm 0.26$ | $\mathbf{95.56 \pm 2.16}$ |
| 500 | $81.72 \pm 1.35$ | 0.45 | $1.4 \pm 0.22$ | $91.72 \pm 3.1$ |

to fit the data well and that the performance does not always improve with more samples. While the accuracy of mRNP reduced less than $1\%$ in very high sample sizes, its performance degraded more than $30\%$ with small sample size of reference set, e.g. 50 samples. Moreover, the model is less robust and shows higher in-distribution entropy in this case. We should also point out that we only explore the effect of reference size, not the quality of the samples. Investigating the automatic selection of reference samples may improve scalability and alleviate the dependence of mRNP on acquiring a reasonable $\mathcal{X}_R$. We leave this for a future study.

**Multi-view CCLE dataset.** We further apply mRNP to explore sensitivity of 472 heterogeneous cell lines to the drug *Panobinostat*. We study the integration of three omics, microRNA sequence (mRNA-seq), reverse phase protein array (RPPA), and metabolites. Following Lee & Schaar (2021), we use ActArea as an indicator of drug sensitivity and then label a response as *sensitive* when this value is higher than median of all responses, and otherwise assign it to the *nonsensitive* group. Note that for TCGA datasets, which we used as o.o.d samples, only two of these modalities are available, i.e. mRNA-seq and RPPA, so we consider metabolite as a missing modality.

The summary of the results are reported in Table 4. mRNP outperforms SOTAs in terms of F1 score while having higher average AUC and AP on the o.o.d. datasets. Lower in-distribution predictive entropy of mRNP compared to DeepIMV and mVAE indicates that mRNP is a better

Table 4: F1 score and uncertainty on CCLE from 100 posterior predictive samples. This is analogous table to Table 1, with different dataset.

| | MOFA | mVAE | DeepIMV | mRNP |
|---|---|---|---|---|
| CCLE | NA / 56.69±5.7 | 0.68 / 63.70±5.30 | 0.67 / 61.54±5.77 | **0.22 / 65.96±4.62** |
| COAD | NA | 0.66 / 49.83/51.41 | 0.62 / 62.93 / 61.84 | **0.41 / 72.47 / 69.67** |
| KIRC | NA | 0.67 / 50.33/51.98 | 0.68 / 67.60 / 64.01 | **0.37 / 68.49 / 65.95** |
| SKCM | NA | 0.67 / 48.64/51.43 | 0.68 / 68.08 / 64.75 | **0.42 / 73.74 / 70.65** |
| Poisson | NA | 0.53 / 23.95/38.81 | 0.67 / 50.35 / 51.76 | **0.64 / 90.72 / 84.24** |
| Average | NA | 0.63±0.0 / 43.18±12.8 / 48.43±6.4 | 0.66±0.0 / 62.24±8.2/ 60.59±6.0 | **0.46±0.1 / 76.35±9.8 / 72.5±8.0** |

choice to deal with high-dimensional data of small sample size, which is often the case in multi-omics data integration when studying complex disease. We further compare the models, based on their robustness to the missing views in the test set. In this ablation study, we train mRNP and two other baselines with training set of CCLE, in which only 3% of the views was missing in the main dataset. At the test time, we artificially introduce missing views by defining a probability of not observing an individual view for each sample. Please note that each view of the samples are treated independently. Then, we increase the probability of views missing from 10% to 50% and measure how different models perform in terms of prediction accuracy. The classification errors of the baselines increase monotonically as the percentage of missing views is increased. mVAE has the worst overall per-

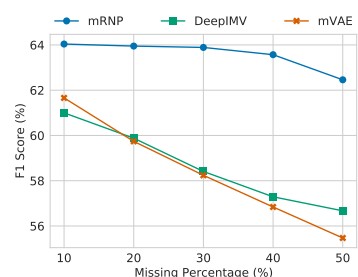

Figure 3: Performance comparison of different methods under various missing probability.

formance, followed by DeepIMV. Using 70% of all the views, the average F1 score of mRNP reduces less than 0.5%, while DeepIMV and mVAE performances degraded around 3%. In the worst case scenario (missing 50% of the views), the mRNP outperforms the DeepIMV and mVAE by 6% and 7%, respectively. This clearly shows the robustness of mRNP to the missing views at test time, and proves that it can faithfully learn individual modalities without any sub-sampling procedure.

## 4.2 ABLATION STUDY TO EXPLORE THE INDUCTIVE BIASES IN MULTI-MODAL REGRESSION

To validate whether mRNP can encode the inductive biases, we consider two toy multi-modal 1-d regression tasks. Following the single-view regression tasks in Louizos et al. (2019) and Osband et al. (2016), we first draw 12 samples from $U[0, 0.6]$ and 8 samples from $U[0.8, 1]$ as the common space between two modalities, and then parameterize the target by $y_i = s_i + \epsilon + \sin(4(s_i + \epsilon)) + \sin(13(s_i + \epsilon))$, where $\epsilon \sim \mathcal{N}(0, 0.03^2)$. Next, we generate observations for the first and second modality as $\mathbf{x}_{v_1} = \mathbf{s}$ and $\mathbf{x}_{v_2} = 6\mathbf{s} + \epsilon$, respectively. Please note that we impose 10% chance that the views are missing. Figure 4 shows the results of both modalities, in which mRNP has a very similar behaviour to the GP. Similar to GP, it reports high uncertainty in the areas where there are no observed samples. For the second regression study, we draw 20 samples from $U[-4, 4]$ for the first modality and simulate the target labels as $y_i = x_{i,v_1}^3 + \epsilon$, where $\epsilon \sim \mathcal{N}(0, 9^2)$. Then, we generate samples for

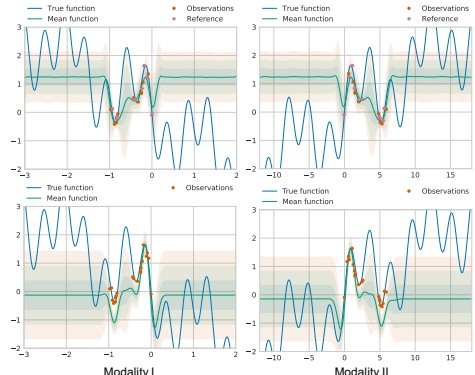

Figure 4: Predictive distributions for the first toy multi-modal regression task based on mRNP (top) and GP (bottom). Shaded areas correspond to $\pm 3$ standard deviations.

the second modality as a nonlinear function $x_{i,v_2} = 2x_{i,v_1} + \eta$ if $x_{i,v_1} \geq 0$ else $0.7x_{i,v_1}$, where $\eta \sim \mathcal{N}(0, 0.1^2)$. Similar to GP, mRNP shows the tendency to quickly move towards a flat prediction outside the areas with observed samples (section D in the supplement).

## 5 CONCLUSION

We have proposed a novel Bayesian relational generative method, tailored for multi-modal neural processes with any missing patterns in their input modalities — mRNP. The mRNP learns the dependency graph among the samples of the given modalities and takes advantage of it to define distributions over the functions of multi-modal data types. Unlike traditional stochastic processes, mRNP can be optimized through mini-batch. By introducing the mixture-of-graphs (MoG), we also could address the issues such as e.g. precision miscalibration of experts and computational complexity observed in the available multi-modal generative models. We have evaluated mRNP on two different real-world multi-modal classification tasks, which demonstrate that not only mRNP substantially outperforms SOTA methods in terms of both prediction accuracy and uncertainty quantification, but also it captures a meaningful joint (structured) representation across views. Furthermore, our visualization results on two toy multi-modal regression tasks show that the predictive distributions of mRNP is similar to a GP with an RBF kernel.

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

## A    DETAILS ON THE EXPERIMENTAL SETUPS, RESOURCES, AND RUNTIME

In both classification tasks, we used early stopping, based on the accuracy on the validation set and no other regularization was employed. In order to avoid sub-optimal initialization in CCLE dataset, we also used the validation set for initialization tuning of the parameters. We also employed a soft-free bits (Chen et al., 2016) modification of the bound to help with the optimization. More precisely, we allowed one free bit on average across all dimensions and batch elements throughout training of mRNP.

During the regression experiments, we randomly select 10 samples from training set as $\mathcal{X}_R$ and consider the dimensions $\{D_u, D_z\}$ as $\{3, 80\}$ and $\{3, 10\}$ for the first and second tasks, respectively. In both cases we use one fully connected layer with ReLu as the activation function to transfer $\mathbf{x}_v$ to $\mathbf{h}_v$, following with a linear layer to learn the parameters of $p_\theta(\mathbf{u}_v \mid \mathbf{x}_v)$, and a linear layer to learn the parameters of $q_\phi(\mathbf{z}_v \mid \mathcal{X})$. The dimension of the hidden layer is set to be 100.

All of the experiments are run on a single GPU node Tesla K80. Each training epoch of mRNP for HW, CCLE, and toy regression took 1.08, 0.53, and 0.02 seconds on average, respectively.

## B    OTHER RELATED WORKS

**Multi-modal VAE (continued).**    Shi et al. (2019) introduces mixture-of-experts multimodal VAE (mmVAE) that does not suffer from potentially overconfident expert. However, it is not able to handle missing modalities during the training, and thus limited in its applicability, especially for biomedical data of small sample sizes with missing views. Our mRNP combines benefits of both mVAE and mmVAE, which is especially critical to deal with high-dimensional data integration. Compared to these approaches, mRNP can in general be more scalable since it does not require sub-sampling as mVAE does. Instead, it effectively takes a vote amongst the graphs and spreads its density over all the individual graph through $\alpha_v$ and $\beta_v$. While the aforementioned models attempt to learn low-dimensional latent variables for multiple views, the focus of mRNP is to introduce distribution over the functions in a supervised setting.

**Conditional VAE (CVAE).**    CVAE (Sohn et al., 2015) are conceptually similar to NPs and has been used to train multi-modal generative models. Unlike the global latent variables that are present in the NPs, the latent variables of CVAEs are local and their decoder functions are deterministic. Hence, it needs to be separately sampled for each of the target predictions $y_i^*$. This is in contrast to the latent variable of an NP that is only sampled once and used to predict multiple values of $y^*$. More precisely, conditioning on the labels of training samples is done by adding the dependency both in the prior $p(z^* \mid y_c)$ and decoder $p(y^* \mid z^*, y_c)$ leading to a deterministic function of the training samples.

**CCA based models.**    There is a rich literature on the extension of CCA to generative modelling (Bach & Jordan, 2005; Virtanen et al., 2011; Klami et al., 2013; Wang et al., 2015; Argelaguet et al., 2018). These extensions can deal with high dimensional data of small sample size (Argelaguet et al., 2018) and outperform classical CCA in terms of interpretability and expressive power. Some of them, such as (Bach & Jordan, 2005; Klami et al., 2013), are generic factor analysis models that decompose the data into shared and view-specific components, and impose an additional constraint to extract the statistical dependencies between the views. Most of the generative methods such as MOFA (Argelaguet et al., 2018) retain the linear nature of CCA while providing more robust

inference compared to the classical solutions. There are also a number of recent models based on VAE that incorporate non-linearity and keep the probabilistic interpretability of CCA (Virtanen et al., 2011; Gundersen et al., 2019). However, they require additional computation to handle missing modalities(Wu & Goodman, 2018).

## C  mRNP AS STOCHASTIC PROCESS

**Proposition I.**    mRNP corresponds to Bayesian models as the distributions defined in Equation (7) are valid, permutation invariant stochastic processes.

*proof.*    Similar to Garnelo et al. (2018b) and Louizos et al. (2019), we rely on de Finetti's and Kolmogorov Extension Theorems Klenke (2013) that states exchangeability and consistency as two sufficient conditions to define a stochastic process.

**Exchangeability.**    This condition requires $p(\mathbf{y}_B \,|\, \mathcal{X}_B)$ to be permutation invariant. More precisely, if each probability density of the model be permutation invariant, then the overall probability will be permutation equivariant. Based on equation (1) in the main paper, we can consider $\bar{\mathbf{u}}_i = f_u(\bar{\mathbf{x}}_i)$, where $\bar{\mathbf{u}}_i = \{\mathbf{u}_{i,v}|v^{\text{th}} \text{ modality present}\}_{v\in\mathcal{V}}$ and $\bar{\mathbf{x}}_i = \{\mathbf{x}_{i,v}|v^{\text{th}} \text{ modality present}\}_{v\in\mathcal{V}}$. Therefore, we can show that the latent variables are permutation equivariant with respect to $\mathcal{X}_B$ as

$$f_u(\sigma(\mathcal{X}_B)) = \sigma(f_u(\mathcal{X}_B)),$$

where $\sigma(\cdot)$ is a permutation function. Then, mRNP defines each elements of the shared graphs $\mathbf{A}$ and $\mathbf{G}$, denoted as $\mathbf{a}_{i,j}$ and $\mathbf{g}_{i,j}$, as a function of $(\bar{\mathbf{u}}_i, \bar{\mathbf{u}}_j)$. Therefore, we also have permutation equivariance for the rows and columns of these two graphs. Based on equation (5) in the main paper, mRNP defines the distribution over $\mathbf{z}_i$ as an independent distribution whose parameters are a function of either $\mathbf{A}$ or $\mathbf{G}$, therefore, the distribution of $\mathbf{z}_i$ is invariant to its parents' permutations. Therefore, the permutation of $\mathcal{X}_B$ will result in the same re-ordering of the $\mathbf{Z}_B$ as follows

$$\sigma(\mathbf{Z}_B) = f_z(\sigma(\mathcal{X}_B)), \tag{7}$$

where $f_z : \mathcal{X}_B \longrightarrow \mathbf{Z}_B$. Let's also define $f_y : \mathbf{z}_i \longrightarrow y_i$. As a result, we can also show $\sigma(\mathbf{y}_B) = f_y(\sigma(\mathcal{X}_B))$.

The product, integral, and summation operators are all permutation invariant. And since we show all random variables of our mRNP model are permutation equivariant to $\mathcal{X}_B$, mRNP model is permutation invariant.

**Consistency.**    This condition requires to show that if we marginalise out a part of $p(\mathbf{y}_B \,|\, \mathcal{X}_B)$, the resulting marginal distribution is the same as that defined on the original space. To this end, we follow Louizos et al. (2019) and define $\tilde{\mathcal{X}}_B = \mathcal{X}_B \cup \{\bar{\mathbf{x}}_0, y_0\}$. Then, we need to show $p(\mathbf{y}_B \,|\, \mathcal{X}_B) = \int p(\mathbf{y}_{\tilde{B}} \,|\, \tilde{\mathcal{X}}_B) \, \mathrm{d}y_0$ in two cases; where $\{\bar{\mathbf{x}}_0, y_0\}$ belongs to either the training set $\mathcal{D}_x$ or the reference set that are not part of the training set, i.e. $\mathcal{X}_R \backslash \mathcal{D}_x$.

In the first case, where $\{\bar{\mathbf{x}}_0, y_0\} \in \mathcal{D}_x$, the new sample will be added to a leaf in the dependency graph. Therefore, it will not affect any of the samples in $\mathcal{X}_R$, hence we can marginalize it out as follow:

$$\int p(\mathbf{y}_{\tilde{B}} \,|\, \tilde{\mathcal{X}}_B) \, \mathrm{d}y_0 = \sum_{\mathbf{G},\mathbf{A},\mathbf{a}_0} \int \prod_{v\in\mathcal{V}} p_\theta(\mathbf{U}_{B,v} \,|\, \mathbf{X}_{B,v}) \, p(\mathbf{G}, \mathbf{A} \,|\, \mathcal{U}_B) \, p_\theta(\mathbf{y}_B, \mathbf{Z}_B \,|\, \mathbf{A}, \mathbf{G}, \mathcal{X}_R)$$

$$p_\theta(\mathbf{u}_{0,v} \,|\, \mathbf{x}_{0,v}) p(\mathbf{a}_0 \,|\, \bar{\mathbf{u}}_0, \mathcal{U}_R) \, p_\theta(\mathbf{z}_0 \,|\, \mathrm{par}_{\mathbf{a}_0}(\mathcal{X}_R, \mathbf{y}_R)) \left( \int p_\theta(y_0 \,|\, \mathbf{z}_0) \mathrm{d}y_0 \right) \, \mathrm{d}\mathcal{U}_B \, \mathrm{d}\mathbf{Z}_B \, \mathrm{d}\bar{\mathbf{u}}_0 \, \mathrm{d}\mathbf{z}_0.$$

As $\int p_\theta(y_0 \,|\, \mathbf{z}_0) \mathrm{d}y_0 = 1$, we can re-write above equation as:

$$\int p(\mathbf{y}_{\tilde{B}} \,|\, \tilde{\mathcal{X}}_B) \, \mathrm{d}y_0 = \sum_{\mathbf{G},\mathbf{A},\mathbf{a}_0} \int \prod_{v\in\mathcal{V}} p_\theta(\mathbf{U}_{B,v} \,|\, \mathbf{X}_{B,v}) \, p(\mathbf{G}, \mathbf{A} \,|\, \mathcal{U}_B) \, p_\theta(\mathbf{y}_B, \mathbf{Z}_B \,|\, \mathbf{A}, \mathbf{G}, \mathcal{X}_R)$$

$$p_\theta(\mathbf{u}_{0,v} \,|\, \mathbf{x}_{0,v}) p(\mathbf{a}_0 \,|\, \bar{\mathbf{u}}_0, \mathcal{U}_R) \left( \int p_\theta(\mathbf{z}_0 \,|\, \mathrm{par}_{\mathbf{a}_0}(\mathcal{X}_R, \mathbf{y}_R)) \, \mathrm{d}\mathbf{z}_0 \right) \, \mathrm{d}\mathcal{U}_B \, \mathrm{d}\mathbf{Z}_B \, \mathrm{d}\bar{\mathbf{u}}_0, \tag{8}$$

where $\int p_\theta(\mathbf{z}_0 \,|\, \mathrm{par}_{\mathbf{a}_0}(\mathcal{X}_R, \mathbf{y}_R))\,\mathrm{d}\mathbf{z}_0 = 1$. We further can summarize the equation (8):

$$\int p(\mathbf{y}_{\tilde{B}} \,|\, \tilde{\mathcal{X}}_B)\,\mathrm{d}y_0 = \sum_{\mathbf{G},\mathbf{A}} \int \prod_{v \in \mathcal{V}} p_\theta(\mathbf{U}_{B,v} \,|\, \mathbf{X}_{B,v})\, p(\mathbf{G}, \mathbf{A} \,|\, \mathcal{U}_B)\, p_\theta(\mathbf{y}_B, \mathbf{Z}_B \,|\, \mathbf{A}, \mathbf{G}, \mathcal{X}_R)$$
$$p_\theta(\mathbf{u}_{0,v} \,|\, \mathbf{x}_{0,v}) \left( \sum_{\mathbf{a}_0} p(\mathbf{a}_0 \,|\, \bar{\mathbf{u}}_0, \mathcal{U}_R) \right) \mathrm{d}\mathcal{U}_B \,\mathrm{d}\mathbf{Z}_B \,\mathrm{d}\bar{\mathbf{u}}_0. \quad (9)$$

Hence, we can further marginalize over $\mathrm{d}\bar{\mathbf{u}}_0$ as:

$$\int p(\mathbf{y}_{\tilde{B}} \,|\, \tilde{\mathcal{X}}_B)\,\mathrm{d}y_0 = \sum_{\mathbf{G},\mathbf{A}} \int \prod_{v \in \mathcal{V}} p_\theta(\mathbf{U}_{B,v} \,|\, \mathbf{X}_{B,v})\, p(\mathbf{G}, \mathbf{A} \,|\, \mathcal{U}_B)\, p_\theta(\mathbf{y}_B, \mathbf{Z}_B \,|\, \mathbf{A}, \mathbf{G}, \mathcal{X}_R)$$
$$\left( \int p_\theta(\mathbf{u}_{0,v} \,|\, \mathbf{x}_{0,v})\mathrm{d}\bar{\mathbf{u}}_0 \right) \mathrm{d}\mathcal{U}_B \,\mathrm{d}\mathbf{Z}_B. \quad (10)$$

Finally, as $\int p_\theta(\mathbf{u}_{0,v} \,|\, \mathbf{x}_{0,v})\mathrm{d}\bar{\mathbf{u}}_0 = 1$, we can drive equation (7) in the main paper. More precisely, we will have

$$\int p(\mathbf{y}_{\tilde{B}} \,|\, \tilde{\mathcal{X}}_B)\,\mathrm{d}y_0 = \sum_{\mathbf{G},\mathbf{A}} \int \prod_{v \in \mathcal{V}} p_\theta(\mathbf{U}_{B,v} \,|\, \mathbf{X}_{B,v})\, p(\mathbf{G}, \mathbf{A} \,|\, \mathcal{U}_B)\, p_\theta(\mathbf{y}_B, \mathbf{Z}_B \,|\, \mathbf{A}, \mathbf{G}, \mathcal{X}_R)\,\mathrm{d}\mathcal{U}_B \,\mathrm{d}\mathbf{Z}_B$$
$$= p(\mathbf{y}_B \,|\, \mathcal{X}_B).$$

In the second care, where $\{\bar{\mathbf{x}}_0, y_0\} \in \mathcal{X}_R \backslash \mathcal{D}_x$, $\{\bar{\mathbf{x}}_0, y_0\}$ belongs to the reference set, but not included in the training set. As we discussed in the main paper, the sets $\mathcal{X}_M$ and $\mathcal{D}_x$ will be same across $\mathcal{X}_B$ and $\tilde{\mathcal{X}}_B$. Hence we need to marginalize out in terms of all samples in $\mathcal{X}_R \backslash \mathcal{D}_x$ as follow

$$\int p(\mathbf{y}_{\tilde{B}} \,|\, \tilde{\mathcal{X}}_B)\,\mathrm{d}y_0 =$$
$$\sum_{\mathbf{G},\mathbf{A}} \int \prod_{v \in \mathcal{V}} p_\theta(\mathbf{U}_{B,v} \,|\, \mathbf{X}_{B,v})\, p(\mathbf{G}, \mathbf{A} \,|\, \mathcal{U}_B)\, p_\theta(\mathbf{y}_B, \mathbf{Z}_B \,|\, \mathbf{A}, \mathbf{G}, \mathcal{X}_R)\,\mathrm{d}\mathcal{U}_B \,\mathrm{d}\mathbf{Z}_B \mathrm{d}y_{i \in \mathcal{X}_R \backslash \mathcal{D}_x}$$
$$= p(\mathbf{y}_B \,|\, \mathcal{X}_B). \quad (11)$$

Therefore, mRNP is consistent under marginalization in both scenarios.

## D  ADDITIONAL EXPERIMENTS

In addition to Figure 4 in the main paper, we further demonstrate that mRNP shows similar behaviour as a GP to encode the inductive biases. In this regression study, we draw 20 samples from $U[-4,4]$ for the first modality and simulate the target labels as $y_i = x_{i,v_1}^3 + \epsilon$, where $\epsilon \sim \mathcal{N}(0, 9^2)$. Then, we generate samples for the second modality as a nonlinear function $x_{i,v_2} = 2x_{i,v_1} + \eta$ if $x_{i,v_1} \geq 0$ else $0.7x_{i,v_1}$, where $\eta \sim \mathcal{N}(0, 0.1^2)$. Figure S1 shows the results of both modalities. Similar to GP, mRNP shows the tendency to quickly move towards a flat prediction outside the areas with observed samples.

For this regression task, we train mRNP for a maximum of 1500 epochs using Adam (Kingma & Ba, 2014) with a learning rate of 0.001. We would also like to point out that we separately train an RBF kernel GP for each modality.

## E  MOG VS POE

As we stated overconfident predictions by one expert in PoE can be detrimental to the whole model. By contrast, MoG does not suffer from potentially overconfident graph, since it effectively takes a vote amongst the graphs, and spreads its density over all the individual graphs. Let us assume that we have two input modalities, and the prediction of the first expert (graph) is noisy and overconfident. We would like to compare the effect of this on two modeling perspectives.

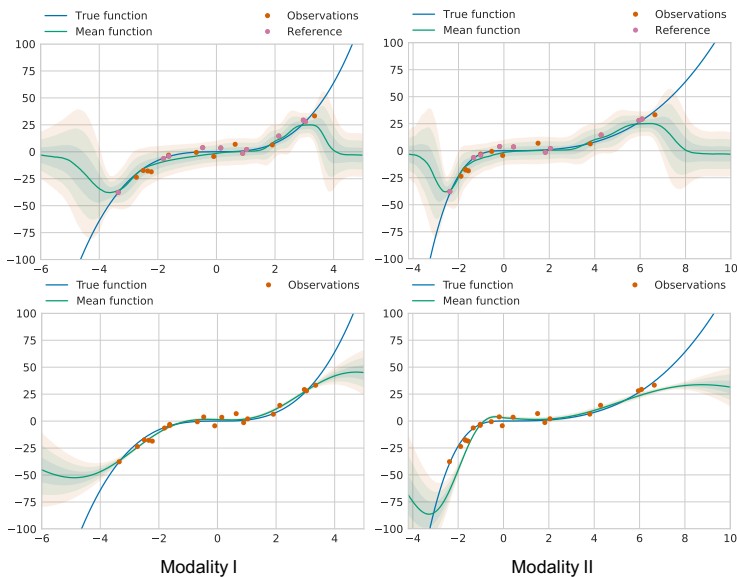

Figure S1: Predictive distributions for the second toy multi-modal regression task based on mRNP (top) and GP (bottom).

**Effect of overconfident predictions of *expert* in PoE model.** PoE assumes $p(\mathbf{z} \,|\, \mathbf{x}_1, \mathbf{x}_2) = p(\mathbf{z} \,|\, \mathbf{x}_1) \times p(\mathbf{z} \,|\, \mathbf{x}_2)$.

Considering we have normal distribution in both experts, i.e. $p(\mathbf{z} \,|\, \mathbf{x}_1) = \mathcal{N}(\mu_0, \sigma_0)$ and $p(\mathbf{z} \,|\, \mathbf{x}_2) = \mathcal{N}(\mu, \sigma_1)$. Therefore, $p(\mathbf{z} \,|\, \mathbf{x}_1, \mathbf{x}_2) = \mathcal{N}(\mu_z, \sigma_z)$, where $\mu_z = \frac{\mu_0 \sigma_1^2 + \mu_1 \sigma_0^2}{\sigma_0^2 + \sigma_1^2}$ and $\sigma_z^2 = \frac{\sigma_0^2 \sigma_1^2}{\sigma_0^2 + \sigma_1^2}$. Let us consider that the first expert is overconfident, i.e. $\sigma_0 \to 0$. As a result, $p(\mathbf{z} \,|\, \mathbf{x}_1, \mathbf{x}_2) = \mathcal{N}(\mu_0, \sigma_0^2)$. This means that the second expert does not matter in this situation and if the first expert be potentially overconfident due to difference in complexity of input modalities or initialisation conditions, second one does not have a chance to correct it.

**Effect of overconfident learning of *graph* in MoG model.** MoG assumes $p(\mathbf{z} \,|\, \mathbf{A}_1, \mathbf{A}_2) = p(\mathbf{z} \,|\, \alpha \mathbf{A}_1 + (1 - \alpha)\mathbf{A}_2)$, where $\mathbf{A}_1$ and $\mathbf{A}_2$ are the graphs in two view.

According to equation (4), $p(\mathbf{z}_i \,|\, \alpha \mathbf{A}_1 + (1 - \alpha)\mathbf{A}_2) = \prod_{k=1}^{\mathcal{D}_z} p(z_{ik} \,|\, \alpha \mathbf{A}_1 + (1 - \alpha)\mathbf{A}_2) = \prod_{k=1}^{\mathcal{D}_z} \mathcal{N}(\mu_{ik}^z, \sigma_{ik}^z)$, where $\mu_{ik}^z = \sum_j (\alpha \mathbf{A}_{1,ij} + (1 - \alpha)\mathbf{A}_{2,ij}) \varphi_k^{\text{prior},\mu}$ and $\sigma_{ik}^z = \exp\left(\sum_j (\alpha \mathbf{A}_{1,ij} + (1 - \alpha)\mathbf{A}_{2,ij}) \varphi_k^{\text{prior},\sigma}\right)$. If we assume $\mathbf{A}_1$ is completely noisy, still $\mathbf{A}_2$ is able to correct it. In addition, we can incorporate prior knowledge through $\alpha$. If we have *a priori* information that the first view is noisy, we can consider $\alpha \ll (1 - \alpha)$ that leads to reduce the negative effect of the first view.

