# OpenReview forum: "Bayesian Relational Generative Model for Scalable Multi-modal Learning"
_ICLR.cc/2022/Conference — ICLR 2022 Submitted_

### Official Review · Reviewer_fVe5 · 2021-10-28

**Correctness:** 2
**Technical Novelty And Significance:** 3
**Empirical Novelty And Significance:** 3
**Recommendation:** 5
**Confidence:** 4

**Main Review:**

**Strengths:** The integration of a dependency graph is a novel contribution to
multimodal learning and overall the empirical results look promising. The paper
is transparent about the limitations of the proposed approach, for instance,
that it is fully supervised and that the results can dependent on the
choice of a good reference set.

**Weaknesses:** First, the paper makes strong claims about previous work that
are not sufficiently well supported. In particular, claims about the
computational efficiency and the improved approximation of the joint posterior
require a better formalization and/or empirical support. Second, the
experiments include mostly unsupervised baselines, but most of the reported
metrics are based on labels, which puts the unsupervised models at a clear
disadvantage. Hence, the experiments would benefit from having more supervised
baselines.

**Computational complexity:**
The paper makes strong claims about the computational inefficiency of previous
approaches (mVAE and DeepIMV), but it is not clear how the proposed method
resolves these issues. Since these arguments are stated very prominently in the
abstract, conclusion, and throughout the paper, a formal comparison of the
computational complexities would be helpful.

**Approximation of the joint posterior:**
The paper claims that the proposed method effectively approximates the joint
posterior of multi-modal data types, in contrast to previous methods.  However,
the unsupervised baselines do not model the same posterior and therefore the
comparison is arguably not very fair. Instead, one could compare to other
supervised or semi-supervised approaches (see separate paragraph).

**Miscalibration of the precision of experts:**
The claim about the miscalibration of experts is positioned very prominently
(e.g., in the abstract and conclusion), but there might not be sufficient
evidence to justify using this explanation to argue in favor of the proposed
approach. I am aware that the claim originates from previous work, but there,
the statement was less central and the results did not depend on this
explanation. However, the present paper seems to convert a hypothetical
explanation to a fact that is heavily relied upon throughout the paper.

**Insufficient supervised and semi-supervised baselines:**
Almost all of the baselines are unsupervised methods, whereas the proposed
approach is fully supervised (a semi-supervised extension is discussed as an
opportunity for future work). It is questionable to frame the requirement of
labeled data as an advantage of the proposed approach compared to existing
multimodal generative models, which are unsupervised and could potentially be
extended to handle semi-supervised data (Joy et al. 2021). Hence, the
experiments seem to be lacking supervised and semi-supervised baselines and the
paper has to explain why it would suffice to consider only DeepIMV as a
supervised baseline.


**Questions and comments:**

- Why do you refer to the proposed approach as a generative model?  Is the
  proposed approach able to *generate* samples from the respective modalities
  modalities? If not, the naming makes the comparison to multimodal generative
  models (e.g., the mVAE) confusing.
- Since you mention CCA-based methods, for which there are many semi-supervised
  and multi-view or multi-modal extensions, did you consider comparing to these
  approaches?
- In the introduction it is written that "training a PoE" is difficult, which
  is confusing. It requires an algorithm to train a model that integrates a
  PoE.
- The authors frequently write that the proposed method "learns individual views
  faithfully". This statement should be made more formal, especially since the
  quality of an encoding is relative to a downstream task.
- In the introduction, it is not clear why stochasticity is a problem for
  multimodal generative models. The statement is not backed up by an
  explanation or reference.
- The relation to attention mechanisms needs to be formalized. In its current
  form, the explanation of the relationship is too superficial.
- Table 2: mRNP accuracy seems to be on a different scale.
- Figure 2: consider using t-SNE or UMAP instead of PCA for the baselines.
- Figure 3: please add values for 0% on the x-axis and standard deviations for
  all methods. Table 4 suggests a significant overlap, if we consider standard
  deviations.


**References**
- Joy, T., Shi, Y., Torr, P. H., Rainforth, T., Schmon, S. M., and Siddharth,
  N.  (2021). Learning Multimodal VAEs through Mutual Supervision. arXiv
  preprint arXiv:2106.12570.


**Summary Of The Paper:**

The paper proposes a Bayesian model that learns a predictive distribution given
multimodal data with labels. The novelty of the proposed method is that it
learns a dependency graph between samples from different modalities and
introduces a new mixture-of-graphs (MoG) method to aggregate different
dependency graphs.  Experimental results on several datasets demonstrate that
the proposed method, which is fully supervised, improves the predictive
performance compared to two unsupervised approaches and one supervised
baseline. Qualitative results corroborate the quality of the learned
representations, which can be visualized as relational graphs, whereas the
baselines require additional dimensionality reduction techniques for
visualization.

**Summary Of The Review:**

The paper introduces an interesting approach to a relevant problem and reports
promising empirical results. However, the paper makes strong claims in its
comparison to previous work, some of which are not sufficiently well supported.
Further, for a fair comparison, the experiments would benefit from more
supervised and semi-supervised baselines. Therefore, I tend towards rejecting
the paper in its current form.

---

> ### Author Response · Authors · 2021-11-13
> **Response to the reviewer fVe5**
>
> We thank the reviewer for their insightful feedback and for finding our method a novel contribution to multimodal learning. Our response is below:
>
> **Computational complexity:** mVAE is using a sub-sampling procedure that includes ELBO terms for whole and partial observations. For instance, with $|\mathcal{V}|$ modalities, we would have $2^{|\mathcal{V}|}$ combinations of inputs. This procedure requires evaluating $2^{|\mathcal{V}|}$ ELBO terms. Since the computational complexity of each individual PoE to calculate precision and Gaussian parameters is $O(N D^2)$, the computational complexity of mVAE will be $O(2^{|\mathcal{V}|} N D^2)$, where $|\mathcal{V}|$ is number of views, $N$ is number of samples, and $D$ is the feature dimension in the embedding space.
>
>
> Unlike mVAE, mRNP only needs to solve one ELBO but needs to infer $|\mathcal{V}|$ bi-partite graphs. Each graph will be learned through multiplication of two embedding matrices $|R| \times D$ and $D \times N$. As a result, the computational complexity of mRNP is $O(|\mathcal{V}|ND|R|)$, where $|R|$ is the number of reference points. Please note that $D$ and $|R|$ are in our control.
>
>
> We further show that even with such a less complex model, mRNP does address what happens when we train the model with a complete dataset, but evaluate it with a test set with missing data. In Figure 3, we trained our model as well as mVAE and DeepIMV with a complete dataset and then artificially imposed missing modalities during the test time. As shown in the paper, the performance of mRNP is more robust to missing samples, and it can learn the individual modalities better than the other baselines in this scenario.
>
>
> **Approximation of the joint posterior:** According to equation (5), the joint posterior distribution of mRNP is $q_{\phi}(Z|X)$, which is not dependent on the input labels and is consistent with the approximate joint posterior of mVAE. Would you please elaborate on this to make sure that we understand it correctly? As we mentioned in the next paragraph following equation (5) of the paper, based on our best knowledge, DeepIMV is the only probabilistic supervised multi-modal method that learns the joint posterior distribution. The reference provided by the reviewer is also under review by ICLR 2022. We are happy to compare our method to it as soon as we could implement their work. Please note that their code is not published yet so we are implementing it from scratch.
>
>
> **Miscalibration of the precision of experts:** Indeed, we provide a mathematical example in Section E in Appendix. The example shows that MoG does not suffer from a potentially overconfident graph whereas overconfident predictions by one expert in PoE can be detrimental to the whole model. We hope that this clarifies the confusion, and if it does not, we appreciate it if you could further elaborate on your concern. We will try our best to resolve/clarify it.
>
>
> **Insufficient supervised and semi-supervised baselines:** mRNP is trying to tackle two main problems in the previous probabilistic multi-modal works, i.e. uncertainty quantification and joint posterior inference. To the best of our knowledge, DeepIMV is the only other approach that can work under the missing view settings and is able to provide uncertainty as well as joint posterior. Thus, we compare the results only with DeepIMV since other methods are unsupervised.
>
> We would also like to emphasize that uncertainty quantification and ood prediction are not necessarily supervised metrics, they rather show the capability of the models when it comes to uncertainty quantification and generalization to the new observations, both of which are critical in unsupervised applications as well. We also show that mRNP is beneficial in terms of providing structured latent space. Lastly, we compared the clustering of embedding space, which is not a supervised metric.

---

> > ### Author Response · Authors · 2021-11-13
> > **Response to the reviewer fVe5 (Part II)**
> >
> > **Comparing with CCA-based model:** MOFA is the SOTA in Bayesian CCA that can handle missing modalities, which we included in the baselines. However, CCA-based models are not able to provide predictive uncertainty.
> >
> > **In the introduction it is written that "training a PoE" is difficult, which is confusing:** We will revise this to make it more informative. As stated in the mVAE paper, PoE does not uniquely specify its component Gaussians.  More specifically, optimizing PoE with a complete dataset with no missing modalities has an unfortunate consequence: since the individual inference networks have not been trained, PoE-based methods do not know how to use them if presented with missing data at test time. Different methods including the sub-sampling procedure have been proposed to address this issue.
> >
> > **Learning individual views faithfully:** This term has been used in different multi-modal generative models, including mVAE, mmVAE, and DeepIMV to address the aforementioned PoE training issue. We will rephrase it in the revised version.
> >
> > **why stochasticity is a problem for multimodal generative models:** As we pointed out in the second sentence in the second paragraph on page 2, stochasticity in the function will help the generative models to be able to better capture uncertainty and improve their generalizability. The footnote on page 2 is also helpful. We will add the following citations Garnelo et al., 2018b;a.
> >
> > **mRNP accuracy seems to be on a different scale:** Thanks to the reviewer for pointing out the typo. The mRNP accuracy is 77.0.
> >
> > **t-SNE instead of PCA for the baselines:** We could see a similar trend using t-SNE. We will add t-SNE plots and put one of them (PCA or t-SNE) into the Appendix.
> >
> > **Relation to attention mechanisms:** We will revise this part to make it clearer. While neural Processes (NPs) propose to define explicit global latent variables in terms of subsets of the data, Attentive NPs [Kim et al, 2019]incorporate attention into NPs, allowing each input location to attend to the relevant context points for the prediction. Similarly, mRNP incorporates cross-attention in the form of a dependency graph among local latent variables and does not define any global latent variable.
> >
> > Kim, H., Mnih, A., Schwarz, J., Garnelo, M., Eslami, A., Rosenbaum, D., ... & Teh, Y. W. (2019). Attentive neural processes. arXiv preprint arXiv:1901.05761.

---

### Official Review · Reviewer_Vnks · 2021-10-31

**Correctness:** 3
**Technical Novelty And Significance:** 2
**Empirical Novelty And Significance:** Not applicable
**Recommendation:** 3
**Confidence:** 3

**Main Review:**

This work proposed a multi-modal relational neural process to provide a predictive distribution. Meanwhile, MoG is introduced to integrate information from different modalities. Generally, this paper is easy to read.

I have the following concerns about the paper:


   1. A literature review is lacking, especially for Gaussian processes-based multi-modal / multi-view classification methods. The discussion of the multi-view Gaussian process method should be conducted. The following papers are very relevant to this paper.

Multiview learning with variational mixtures of Gaussian processes. Knowl. Based Syst. 2020

Multi-view representation learning with deep gaussian processes.

Multimodal similarity gaussian process latent variable model.


2. Some statements in the article are biased. For example, the introduction states that “existing generative models for multi-modal learning focus on latent representation, but do not fully incorporate the label information.”  Generative models are only one of the branches of multimodal learning, and many multimodal supervised models have been developed, e.g., DeepIMV in the paper.
The proposed MoG is not convincing and novel, which just averages the graphs from different modalities. In my opinion, this contribution is weak.

3. The experiment of the paper is not sufficient. The paper only compared the two baseline methods on two real-world datasets. At the same time, the experiment did not use large-scale datasets to illustrate the author’s claim "makes mRNP scalable to large datasets through mini-batch optimization".

In addition, I found some problems in the writing.
Incorrect use of semicolon “;” and colon “:” in the introduction.
The meaning of the symbol is not introduced when it first appears (e.g., φ$_{sim}$ in Eq 2).
The title of the paper is multi-modal, but multi-view is used many times in the paper.

**Summary Of The Paper:**

This paper proposes a multi-modal Relational Neural Process, which learns a dependency structure among the samples.  To integrate information from multiple modalities, mixture-of-graphs (MoG) is introduced.

**Summary Of The Review:**

The article should conduct more thorough literature research, method design, and improvement of writing.

---

> ### Author Response · Authors · 2021-11-13
> **Response to the reviewer Vnks**
>
> **Literature review:** Thanks to the reviewer for pointing this out. We will add the following paragraph to the section of the related work.
>
> [multi-view Gaussian Processes] In order to increase the expressive power of CCA-based methods, Song et al. proposed non-parametric mapping functions in order to transform heterogeneous modalities into a shared latent space. More recently, MvMGPs [Sun et al] proposed a supervised multi-view learning approach based on the variational mixture of GPs. MvMGPs learn an embedding space for each individual modality and add a Kullback-Leibler regularization to minimize the divergence between the posterior distributions of latent spaces in two views. However, this model is not able to learn a joint embedding space. [Sun et al] proposed MvDGPs to better exploit the characteristics of multi-view representation learning and deep Gaussian processes. It extends DGPs to multi-view representation learning so that they can process multi-view data and improve the performance of machine learning tasks. While these models are able to provide stochasticity over the functions, they cannot handle missing modalities similar to mRNP.
>
> **Generative models are only one of the branches of multimodal learning, and many multimodal supervised models have been developed, e.g., DeepIMV in the paper.** We will revise it to “most of the probabilistic models”.
>
> **The proposed MoG is not convincing and novel, which just averages the graphs from different modalities.** Although we chose to use mean aggregation over the graph, we can use any other aggregation method here. We think that simpler approaches work better in practice and are more robust. Similarly, mVAE, mmVAE, and DeepIMV have contributions that can be considered simple such as using multiplication/summation of posterior distributions in VAE setting, but we think that they are valuable and are glad that they are published.
>
> The key point in our work is that our approach is principled, and we provide mathematical proof that it can handle overconfident views in a multi-view setting. Empirically, we show that this is the first model that can 1) provide a structured representation learning, 2) outperform the existing baselines with a large margin in terms of uncertainty quantifications and ood detection, 3) capture inductive biases similar to RBF Kernels in GPs.
>
> **Concern regarding experiments:** To the best of our knowledge, we compare our method to the SOTA multi-modal probabilistic models that can provide joint posterior distribution in the setting of multi-modal input with arbitrary missing views. We would be happy to compare our method with any probabilistic supervised multi-modal method for missing view setting if the reviewer can kindly provide us with relevant references.
>
> In addition, we show that ELBO in our framework can be computed through a mini-batch. As instructed, we further evaluate our model on the TCGA dataset and compare its performance with DeepIMV. We perform a 3-year mortality classification based on the comprehensive observations from two omics data on 7295 cancer cell lines. Data includes two different modalities; microRNA expression and reverse-phase protein array (RPPA). The classification accuracies are 60.44% and 56.96% for mRNP and DeepIMV, respectively. We used 300 samples as the reference samples throughout this experiment.

---

> > ### Comment · Reviewer_Vnks · 2021-11-29
> > **Thank you for your response**
> >
> > After carefully reading your response  and the comments of other reviewers, I decided to keep my opinion unchanged. The methods and experiments in this paper still need to be improved. For example, almost all reviewers mentioned that the experiment was inadequate. At the same time, both the reviewer bvzL and I were confused about the method of the paper.

---

### Official Review · Reviewer_VW19 · 2021-10-31

**Correctness:** 3
**Technical Novelty And Significance:** 2
**Empirical Novelty And Significance:** 2
**Recommendation:** 5
**Confidence:** 5

**Main Review:**

The method is interesting, and the introduction is attractive to read the following. But the part for methods are a little confusing. Moreover, it seems the authors are not clear about variational inference. The paper in page 5 shows Evidence Lower Bound (ELBO) is minimized. In fact, it should be maximized.

**Summary Of The Paper:**

This paper tries to use Bayesian relational generative model for scalable multi-modal learning. They propose a class of stochastic processes that learns a graph of dependencies between samples across multi-modal data types through adopting priors over the relational structure of the given data modalities. The so-called mRNP method can address the limitations in joint posterior approximation.

**Summary Of The Review:**

It may not be so fit for iclr.

---

> ### Author Response · Authors · 2021-11-13
> **Response to the reviewer VW19**
>
> Thanks to the reviewer for pointing out the typo. We will fix it in the revised version. We would also appreciate it if the reviewer could elaborate on things that are currently lacking in the paper and that can improve the paper. We have spent a considerable amount of time on the provided paper. Although it might be time-consuming, we would like to kindly ask the reviewer to share more detailed comments on our submission. It is important for us to know which parts of the paper need more clarification. We might have time to revise it and make it more clear during this period.

---

### Official Review · Reviewer_bvzL · 2021-11-02

**Correctness:** 3
**Technical Novelty And Significance:** 2
**Empirical Novelty And Significance:** Not applicable
**Recommendation:** 3
**Confidence:** 4

**Main Review:**

Strength:

- The problem raised in this paper is interesting. Due to the presence of missing modalities at test time, estimating the uncertainty is important in practice.

- While the neural process has been applied in a wide range of ML tasks, it is the first time to be applied to multimodal learning.


Weakness:

- The writing of this paper is terrible. It is difficult to understand to the main content of this paper by simply going through the paper from top to bottom. I spend lots of time to figure out what’s the main idea of this paper, but failed. Then, I came to one of the reference of this paper, Functional Neural Process, then everything becomes clear. The idea in Functional Neural Process is introduced much better and clearer than this submission.

- This submission looks strikingly similar to the paper of Functional Neural Process. I can say that the underlying ideas of the two paper are almost the same, including the introduction of directed acyclic graph in the model. A slight improvement is to mix multiple graph into one (MoG) to handle the modality missing problem. However, the content relevant to this part only account for a very small proportion. Most of the techniques used in the paper have been discussed in the previous FNP paper. It is better to pay more attention to how to adapt FNP on multimodal learning. Besides, how does the performance look like with FNP in figure 4 and s1?


- To address modality missing problems, the authors propose a MOG method to model the relationship among different modalities. However, the construction of graphs A and G exactly confuses me. You first construct A_v and G_v for each modality separately, and then the whole graph A and G is the sum of all subgraphs (A_v, G_v) on different modalities. That means, graphs A and G are shaping in a block-like, and there are no edges between different views. From this perspective, there are no interactions among different modalities. The whole training process is equal to the way that trains multi models on different modalities separately, but they share the same parameters. This is a quite different training scheme compared with previous multimodal learning models, and also confusing.

- Uncertainty estimation is an important issue in multimodal learning. Ideally, the more modalities missed at test time, the higher uncertainty it is. However, there are no corresponding experiments to evaluate this hypothesis. I think it is necessary to explore how does the missing rate at test time influence the uncertainty degrees.

- Another question is how do you evaluate the model. If I understand correctly, given a training data {\bar{x}, y}, you assign each data point x_v \in \bar{x} the same label, i.e., y, and use it to maximize the ELBO. At test time, according to the predictive distribution, you predict a label y_v for each test data x_v \in \bar{x}. Since there is a unique label for \bar{x}, my question is, how do you predict the label for \bar{x}, according to the predictions y_v on different views? Maybe I miss something, but I think it needs to be clarified more clear.


**Summary Of The Paper:**

This paper proposed to use neural processes for supervised multi-modal learning, which have the abilities to estimate the uncertainty of prediction and to handle missing modalities. Specifically, a directed acyclic graph is learned for each modality in the neural process, which are then used to construct a mixture-of-graphs (MoG) to sidestep the modalities missing problem. The introduction of inducing points makes the predictive distribution tractable. Experimentally, on the label prediction and uncertainty estimation tasks, the proposed model performs well compared to the recent multimodal learning methods.

**Summary Of The Review:**

The problem investigated in this paper is. But I think the contribution of this paper is not clear and the writing of this paper should be improved. I recommend the authors to take a thorough review of the field of neural process and think about how effectively adapt the it to address the modalities missing problems.

---

> ### Author Response · Authors · 2021-11-13
> **Response to the reviewer bvzL**
>
> **Reviewer assumes graphs A and G are shaping in a block-like, and there are no edges between different views. Based on this assumption, there are no interactions among different modalities. As a result, the reviewer argues the whole training process is equal to the way that trains multi models on different modalities separately, but they share the same parameters which is a quite different training scheme compared with previous multimodal learning models.** Let us clarify a point of confusion. As shown in Figure1, we are generating a graph of dependencies between samples in each view. As a result, we have two bipartite graphs in each view, i.e. $A_V (|R| \times N)$ and $G_V (|R| \times |R|)$, and their nodes in different views are the same and the only difference is their edges. So, we are combining them through MoG. Therefore, the model is an end-to-end model. We do not have any parameter sharing in the encoder part. We argue the training procedure is similar to the previous approaches. Please let us know if this is still not clear so we can clarify it further.
>
> **It is necessary to explore how does the missing rate at test time influence the uncertainty degrees:** We thank the reviewer for this suggestion. Indeed, we found it very interesting and conducted an ablation study based on this. We will add it as a plot into the revised paper, but here are the results:
>
> We increase the missing modality from 0 to 50% for HW dataset. The predictive entropy increases with more missing modalities as expected:
>
> | Missing Percentage (%) 	| 0    	| 10   	| 20   	| 30   	| 40   	| 50   	|
> |------------------------	|------	|------	|------	|------	|------	|------	|
> | Predictive Entropy     	| 0.40 	| 0.58 	| 1.26 	| 1.46 	| 1.51 	| 1.76 	|
>
> **How do we evaluate the model:** We believe there is a point of confusion here as well. First, we split each dataset to train, validation, and test sets and each sample (including all of its available modalities) will be included in one of these splits. Second, we maximize ELBO using the training set. Third, we put each $\bar{x}$ into the encoder during the test and predict one $y$ for each sample (not each modality).
>
> More specifically, we do not assign label $y$ to each view separately. Rather, during the training of the parameters of decoder $\theta$ and encoders $\phi$, we first embed each domain to a domain-specific latent space and then construct a graph of dependency for each view. Next, we combine the graphs to create a single graph between samples in the dataset, and use the single graph to learn a shared latent space $Z$, and use $Z$ to predict $y$. Please note that 1) we only use the shared latent space to generate one label for multiple views. Therefore, we only use the label for all views in only one likelihood. 2) We only use the label of reference points to construct the prior distribution of $Z$. Again, we only use one label for all views in this case.
>
> At test time, we only use new test samples $\bar{x}^*$ and put them into the encoder without any label. Please note that we do not use any view from the test samples during training. We construct view-specific latent space u* and then a bi-partite graph between reference samples and the test sample. After that, we can construct the shared latent space and then put that $Z$ to the decoder with parameter $\theta$ to get $y^*$. As a result, we are not using any label for any of the modalities during the test.
>
> **Underlying ideas of the two papers are almost the same, including the introduction of directed acyclic graph in the mode.** First, right after equation (2), we cite FNP and mention that we learn the directed acyclic graph similar to FNP.  Second, the novelty of the paper is not the way we learn the graph, but rather the novelty of our work is learning the graph and combining them to come up with a shared latent space. Third, in the related works, we wrote “Closely related to our mRNP, Louizos et al. (2019) discards the idea of the global latent variables and instead builds a graph of dependencies among local latent variables, making it more suitable for modeling high-dimensional data. […] However, none of these works addresses multi-modal learning problem”. Please also note that the directed acyclic graph is not originated from FNP and is known for a long time.
>
> We would like to point out that the problem that we are tackling is learning the joint posterior distributions and uncertainty quantification in a multi-modal setting with arbitrary missing patterns. To the best of our knowledge, our work is the first one solving it through a graph learning procedure.

---

> > ### Author Response · Authors · 2021-11-13
> > **Response to the reviewer bvzL (Part II)**
> >
> > **Improvement is to mix multiple graphs into one (MoG) to handle the modality missing problem.** FNP is not a multi-modal learning method and is not able to tackle a similar problem. Please note that we propose to learn a graph of dependency in each view and introduce MoG to combine them to provide a shared latent space. The previous methods used PoE to multiply the posterior distribution of multiple views. Apart from that, FNP is tackling a completely different problem and is not able to solve multi-modal learning, even with complete data. In addition, we have theoretically shown that the proposed method is the first NP for a multi-modal setting, and can address the potential problems of previous probabilistic multi-modal learning methods.
> >
> >
> > **How does the performance look like with FNP in figure 4 and s1?** FNP is not able to handle multi-modal learning. Could you please elaborate on how we can compare with that?
> >
> > **The writing of this paper is terrible.** We would appreciate it if the reviewer can int us to the point of confusion in the framework shown in Figure 1 so that we can make it more clear. In the following, we will try to describe the model in plain language and then point out our main contributions, hopefully, make the understanding of the paper much better.
> >
> > In this paper, the main goal is to learn a joint posterior distribution as well as to capture uncertainty for multi-modal inputs with any arbitrary patterns under a supervised setting. To do so, 1) we first embed each view to a view-specific latent space; 2) we construct a graph of dependencies between different samples in each view; 3) combine the learned graph through MoG to construct a single bi-partite graph; 4) compute the distribution of the shared latent space Z using the inferred graph; 5) construct the label of the sample by using Z as input to the decoder.
> >
> > Steps 1-4 can be considered as encoding data in VAE setting and the last step as the decoder.
> >
> > Our main contributions are: 1) We develop a novel multi-modal Relational Neural Process, mRNP, that defines a distribution over functions for multi-modal data by employing local latent variables, and learns a dependency structure among the samples of the given modalities. 2) We theoretically prove the exchangeability and consistency of mRNP for a multi-modal setting, two necessary conditions that have to be satisfied during the construction of such a model. Thus, we show that mRNP is a valid stochastic process. 3) We further show that the local latent variable structure in mRNP is able to encode inductive biases and demonstrate this by designing an mRNP model that behaves similarly to a GP with an RBF kernel (an ablation study). 4) We introduce mixture-of-graphs (MoG) in our model construction that can address the issues in multi-modal learning.

---

### Decision · Program_Chairs · 2022-01-20

**Decision:**

Reject

**Comment:**

The paper extends the FNP model to multimodal settings using the mixture of graphs. However, there are legitimate concerns about the quality of experiments, such as baselines, as the reviewers mention. For example, mRNP is supervised, and comparison to DeepIMV is not fair. I encourage the authors to address them appropriately in the next version of the paper.

The authors can significantly improve the presentation of ideas. Please avoid making hyperbole and excessively bold statements, as the reviewers have pointed out. This way, there will be room for a better demonstration of the novel parts of the paper. For example, the authors misuse the term "generative" for the proposed mRNP. There are multiple hand-waving statements about the role of uncertainty that are not well-supported in the current draft. I believe this paper can be a good paper by addressing the reviewers' comments.